# Tree-Ring Stable Oxygen Isotope Ratio ($\delta^{18}$O) Records Precipitation Changes over the past Century in the Central Part of Eastern China

Changfeng Sun [1,2], Xuan Wu [1], Qiang Li [1,2,*], Yu Liu [2,3,*], Meng Ren [2,4], Qiufang Cai [2,3], Huiming Song [1,2] and Yongyong Ma [5]

1   Institute of Global Environmental Change, Xi'an Jiaotong University, Xi'an 710049, China; sunchangfeng@xjtu.edu.cn (C.S.)
2   State Key Laboratory of Loess and Quaternary Geology, Institute of Earth Environment, Chinese Academy of Sciences, Xi'an 710061, China
3   CAS Center for Excellence in Quaternary Science and Global Change, Chinese Academy of Sciences, Xi'an 710061, China
4   Xi'an Institute for Innovative Earth Environment Research, Xi'an 710061, China
5   Meteorological Institute of Shaanxi Province, Xi'an 710015, China; ma-yong1990@163.com
*   Correspondence: liqiang@xjtu.edu.cn (Q.L.); liuyu@loess.llqg.ac.cn (Y.L.)

**Abstract:** Fully understanding the past characteristics of climate and patterns of climate change can contribute to future climate prediction. Tree-ring stable oxygen isotope ratio ($\delta^{18}$O) is crucial for high-resolution research of past climate changes and their driving mechanisms. A tree-ring $\delta^{18}$O chronology from 1896 to 2019 was established using *Pinus tabulaeformis* Carr. from the Yimeng Mountains (YMMs) in the central part of eastern China. We found that precipitation from the 41st pentad (five days) of the previous year to the 40th pentad of the current year ($P_{41-40}$) was the main factor influencing the YMMs tree-ring $\delta^{18}$O change. We then created a transfer function between $P_{41-40}$ and tree-ring $\delta^{18}$O. The reconstructed $P_{41-40}$ explained 39% of the variance in the observed precipitation during the common period of 1960–2016. Over the past 124 years, the YMMs experienced 19 dry years and 20 wet years. The spatial correlation results indicate that the reconstructed precipitation could, to some extent, represent the precipitation changes in Shandong Province, and even the central part of eastern China, from the early 20th century to the present. In addition, it was found that the trends in YMMs tree-ring $\delta^{18}$O were similar at both high frequency and low frequency to those in tree-ring $\delta^{18}$O series from Mt. Tianmu in eastern China and from Jirisan National Park in southern South Korea. However, the YMMs tree-ring $\delta^{18}$O was only correlated at low frequency with the tree-ring $\delta^{18}$O of the Ordos Plateau in northwestern China and that of Nagano and Shiga in central Japan, which are far from the YMMs. The changes in precipitation and tree-ring $\delta^{18}$O in the YMMs were, to some extent, influenced by the Pacific decadal oscillation.

**Keywords:** tree-ring stable oxygen isotope; precipitation reconstruction; Yimeng Mountains; eastern China; Pacific decadal oscillation

## 1. Introduction

Climate changes, especially extreme climate events, have significant impacts on human activities and economic development. Understanding the change characteristics and driving mechanisms of climate can help predict its future changes and prevent extreme climate disasters [1]. However, due to their limited duration, observational data fall short in providing a comprehensive understanding of the characteristics of climate change. Research into past climate, especially high-resolution climate reconstruction, can compensate for this deficiency [2]. Tree rings have been widely used in global paleoclimate change studies due to several advantages, including their long sequences, accurate cross-dating, continuity, high resolution and high sensitivity to climate change, and are considered an important

technique for obtaining high-resolution data on past climate change [3–7]. Furthermore, tree rings are essential in predicting future climates [8]. Tree-ring stable oxygen isotope ratio ($\delta^{18}$O) has become an important indicator for annual resolution paleoclimate reconstruction and atmospheric circulation research due to multiple advantages, such as small sample sizes, a clearly interpreted physiological mechanism, a lack of need for detrending, and less influence from age effects, as well as retention of more low- and high-frequency climate signals [9–12]. Early tree-ring $\delta^{18}$O research was conducted in Europe and America, where a relatively complete data network of tree-ring $\delta^{18}$O has been established and significant results have been achieved [13–15]. At present, there are few studies on tree-ring $\delta^{18}$O in China, and they are mainly concentrated in western China. For example, tree-ring $\delta^{18}$O of Qilian Juniper in the northeastern Tibetan Plateau region indicated changes in relative humidity over the last thousand years [16]. Additionally, there was a modest drying trend during the Middle Ages Climate Anomaly, a noticeable wet phase during the majority of the Little Ice Age, and increasing moisture associated with the twentieth century warming trend. Based on tree-ring $\delta^{18}$O from multiple regions, it was found that the primary cause of the drought in the southern Tibetan Plateau since the 1860s could be the weakening of the Asian summer monsoon [17]. Compared with central and western China, there are fewer studies on tree-ring $\delta^{18}$O in eastern China. Using tree-ring $\delta^{18}$O of *Fokienia hodginsii* (Dunn) Henry et Thomas, precipitation variations in Fujian Province over the past 100 years were reconstructed [12], and it was found that diminished El Niño-Southern Oscillation (ENSO) variance led to a weakened correlation between tree-ring $\delta^{18}$O in Fujian and the sea surface temperatures (SST) of the tropical Pacific. Based on tree-ring $\delta^{18}$O, the Palmer Drought Severity Index (PDSI) in the Mt. Tianmu region during the period of 1618–2013 was reconstructed, and it was found that tree-ring $\delta^{18}$O was connected to meiyu rainbelt precipitation and the East Asian summer monsoon (EASM) intensity during the May–June period [18].

The Yimeng Mountains (YMMs) are situated in the region of eastern China and are also an important component of the Tai-Yi Mountain Range in Shandong Province [19]. The YMMs belong to the temperate monsoon climate zone, which is influenced by the Asian monsoon and is vulnerable to global change. Climate change plays a crucial role in shaping the ecological environment of the YMMs and extends their impact to the broader eastern monsoon area of China. In addition, the population of the YMMs and the surrounding region is dense, and climate change plays a crucial role in the agricultural and economic development. Understanding past climate change has instructive significance for the economic and social development, ecological environment protection, and other aspects of the region [20]. However, to date, the changing patterns of climate in the YMMs and even the Shandong Province are still unclear, with especially limited understanding of the characteristics of past centuries' hydroclimate change. Therefore, it is necessary to conduct high-resolution climate reconstruction research on the YMMs region in order to fully understand climate variations in the Shandong Province and eastern China and to predict future climate. Moreover, the research to date on dendroclimatology in the YMMs and the central part of eastern China mainly focuses on tree-ring width and carbon isotopes [20–22], and there has been no research on tree-ring $\delta^{18}$O.

In this study, we selected *Pinus tabulaeformis* Carr. in the YMMs as the research object and determined the tree-ring $\delta^{18}$O sequence after accurate dating of the rings. Firstly, this study establishes the tree-ring $\delta^{18}$O series of the YMMs, and then reconstructs the precipitation changes over the previous 120 years in order to analyze its features and patterns of change. Finally, we will discuss the impact of driving factors such as the Asian monsoon and the Pacific decadal oscillation (PDO) on the long-term precipitation changes in the YMMs region.

## 2. Materials and Methods

### 2.1. Study Region and Sampling Site

The YMMs are located in Linyi, Shandong Province in the central part of eastern China (Figure 1). The annual precipitation is 717 mm, with the bulk of precipitation occurring predominantly in July and August (Figure 2). The minimum precipitation in January is less than 10 mm, while the maximum precipitation in July is approximately 200 mm. The mean annual temperature is 12.7 °C, with the highest recorded temperature of 25.9 °C in July and the lowest recorded temperature of −2.4 °C in January. The monthly difference in relative humidity is not significant, ranging from 56% to 80%. For this research, *Pinus tabuliformis* was selected in the sampling site (YMMs, 35°36′ N, 117°51′ E; 1048–1100 m above sea level; Figure 1), and 98 cores were collected from 49 trees. The growing season of *Pinus tabulaeformis* in the YMMs is from April to October which is also the formation time of tree rings.

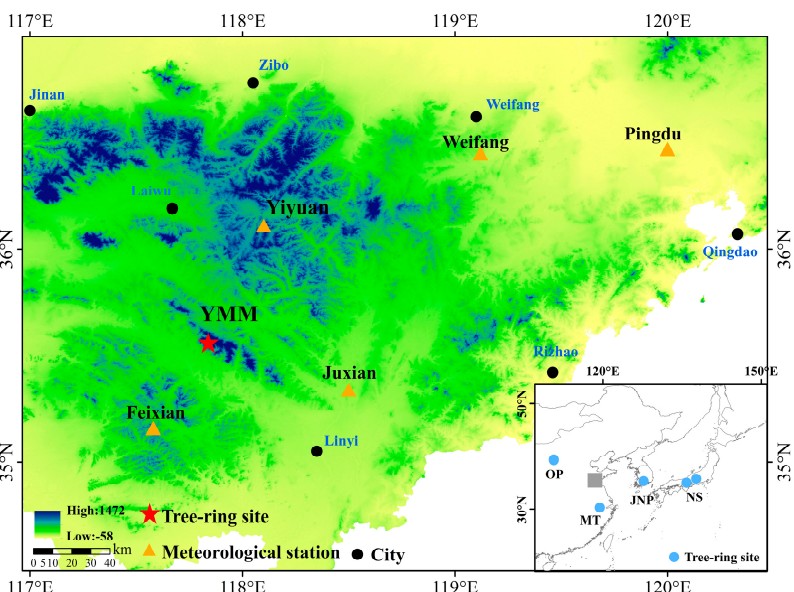

**Figure 1.** This map shows the locations of the meteorological stations and the tree-ring sampling site. The gray rectangles and blue circles in the inset in the bottom-right indicate the sites used for comparisons (OP: Ordos Plateau in northwestern China; MT: Mt. Tianmu in eastern China; JNP: Jirisan National Park in southern South Korea; NS: Nagano and Shiga in central Japan).

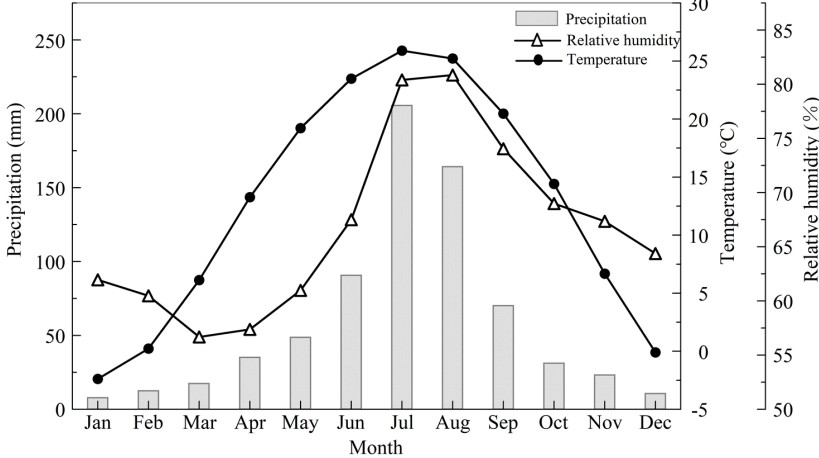

**Figure 2.** The mean monthly precipitation (P), temperature (T), and relative humidity (RH) of the study region based on five meteorological stations during the period 1959–2016.

### 2.2. Tree-Ring $\delta^{18}O$ Measurement

Utilizing dendrochronological methods, each tree-ring core was visually dated and subsequently measured with the LINTAB measurement system to a precision of 0.01 mm. The COFECHA program was instrumental in maintaining the quality of cross-dating [23], providing an effective means for identifying and excluding any potential false or missing rings. Any cores which were fractured and those for which a specific calendar year could not be determined were excluded. Finally, 92 cores from 47 trees were retained for analysis. The COFECHA results showed that the average of correlation coefficient among these 92 individual series was 0.48 and that the mean sensitivity was 0.35. After determining the calendar year, five cores with clear annual rings and fewer missing rings were chosen, namely, YS1-12B, YS2-24A, YS2-02B, YS1-22A, and YS2-08A (Table 1). Under microscopic examination, the rings of these five cores were sequentially peeled off. The extraction of $\alpha$-cellulose was conducted using the enhanced Jayme–Wise method [24]. The obtained $\alpha$-cellulose was homogenized with an ultrasonic cell disruptor and freeze-dried. Approximately 0.12–0.16 mg of eligible samples was enclosed within silver capsules and these were placed in a Delta V Advantage elemental mass spectrometer to obtain oxygen isotope values. The oxygen isotope ratio was denoted as $\delta^{18}O$, representing its deviation from the Vienna standard mean ocean water (VSMOW). In our laboratory, we selected MERCK microcrystalline cellulose (27.7‰) as the standard. Through repeated measurements of the working standard, we ensured a measurement accuracy of less than $\pm 0.2$‰. The composite tree-ring $\delta^{18}O$ chronology was derived by calculating the arithmetic mean of these five series through the numerical mix method [18].

**Table 1.** Statistical features of five individual series as well as the composite series.

| Statistical Parameters | YS1-12B | YS2-24A | YS2-02B | YS1-22A | YS2-08A | Composite |
|---|---|---|---|---|---|---|
| Length (years) * | 89 | 102 | 103 | 108 | 102 | 124 |
| Start year | 1916 | 1917 | 1916 | 1896 | 1917 | 1896 |
| End year | 2008 | 2019 | 2019 | 2007 | 2019 | 2019 |
| Maximum value (‰) | 32.72 | 31.19 | 32.65 | 30.66 | 34.81 | 32.23 |
| Minimum value (‰) | 26.42 | 24.27 | 25.79 | 24.42 | 24.73 | 26.01 |
| Mean value (‰) | 29.22 | 27.91 | 28.98 | 28.20 | 29.32 | 28.67 |
| AR1 | 0.16 | 0.25 | 0.33 | 0.32 | 0.13 | 0.20 |
| Standard deviation (‰) | 1.34 | 1.37 | 1.32 | 1.11 | 1.49 | 1.07 |
| Skewness | 0.06 | 0.07 | 0.16 | 0.34 | 1.58 | 0.37 |
| Kurtosis | 0.23 | −0.15 | 0.18 | −0.30 | 0.34 | 0.18 |

* It should be noted that some cores contain missing rings, so the length is not consistent with the difference between the end year and start year. AR1: first-order autocorrelation [25].

### 2.3. Meteorological Data

Tree-ring $\delta^{18}O$ mainly reflects regional climate change; therefore, five meteorological stations were chosen in close proximity to the sampling site, namely, Yiyuan, Weifang, Juxian, Pingdu, and Feixian (Figure 1). Due to the significant correlation between their precipitation, temperature, and relative humidity (RH), the average data of the five stations were used to represent regional climate change (Figure 2). The meteorological data mainly come from the National Meteorological Science Data Center. In addition, the gridded data utilized in this study were obtained from the Climatic Research Unit (CRU) [26]. The other tree-ring $\delta^{18}O$ sequences come from the National Centers for Environmental Prediction (NCEP) [27], and the PDO data were sourced from the KNMI Climate Explorer (https://climexp.knmi.nl/start.cgi, accessed on 23 December 2023).

### 2.4. Methods

Pearson correlation analysis was used to investigate the relationship between tree-ring $\delta^{18}O$ and climate factors. Additionally, a linear regression model based on the least squares method was utilized to recreate historical precipitation patterns. The assessment

of the reconstruction model's dependability and stability was conducted using the split calibration-verification approach [28]. The statistical parameters utilized to evaluate the calibration period included the Pearson correlation coefficient (r) and the explained variance (R2). In order to assess the accuracy of the calibrations, additional measures such as r, R2, the reduction of error (RE), the coefficient of efficiency (CE) and the sign test (ST) were employed. The values of RE and CE need to be higher than zero in order for the model to be accepted [29]. Using the KNMI Climate Explorer, spatial correlation analysis was applied to describe the variability in the climate signal at the regional scale. A low-pass filter was utilized, which accepts signals below a cutoff frequency but attenuates frequencies over the cutoff frequency [30] The filter creates gradual changes in the output values by eliminating multiple frequencies, which facilitates the identification of low-frequency properties.

## 3. Results and Discussion

### 3.1. Tree-Ring $\delta^{18}O$ Chronology

The mean values of the five individual tree-ring $\delta^{18}O$ series in the YMMs range from 27.91‰ to 29.32‰, and the correlation among these five series is very significant (Table 2). We used the numerical mixing method to synthesize the five series into a master series sequence (Table 2, Figure 3). The mean $\delta^{18}O$ of the composite series from 1896 to 2019 was 28.67‰, with a standard deviation of 1.07‰.

**Table 2.** Correlation coefficients among the five individual $\delta^{18}O$ series.

|  | **YS1-12B** | **YS2-24A** | **YS2-02B** | **YS1-22A** |
|---|---|---|---|---|
| YS2-24A | 0.753, 88 |  |  |  |
| YS2-02B | 0.550, 88 | 0.419, 101 |  |  |
| YS1-22A | 0.708, 83 | 0.745, 86 | 0.418, 87 |  |
| YS2-08A | 0.506, 87 | 0.410, 100 | 0.575, 100 | 0.531, 83 |

Note. The correlation coefficients in the table all passed the significance test.

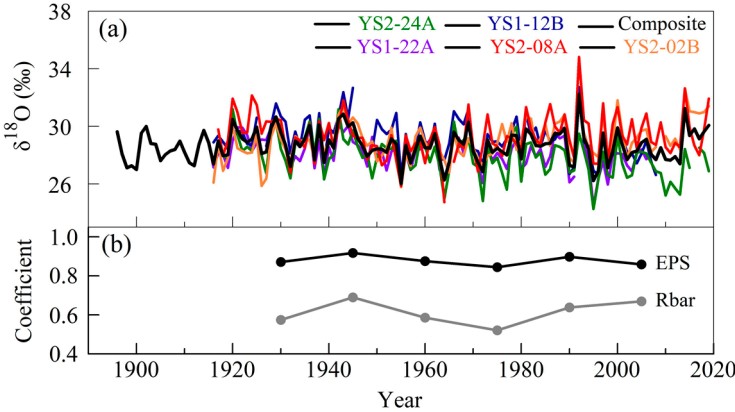

**Figure 3.** (**a**) Five individual series and the composite series of $\delta^{18}O$ created by the numerical mixing method. (**b**) The running mean interseries correlation (Rbar) and expressed population signal (EPS) calculated using 30-year windows with a lag time of 15 years.

The expressed population signal (EPS) and Rbar (the mean correlation coefficient of individual series) were used to determine the effective length of the tree-ring chronology, and an EPS value greater than 0.85 was considered the threshold for the beginning year of the tree-ring $\delta^{18}O$ chronology. Since 1916, there have been three $\delta^{18}O$ sequences with EPS values greater than 0.85 (Figure 3). Due to the significant correlation between the core YS1-22A and other sequences during their common time period (1916–2007), the composite series during 1896–1915 containing only the core YS1-22A could also be used to reflect regional tree-ring $\delta^{18}O$ changes.

### 3.2. Responses of Tree-Ring δ¹⁸O to Climate

The response analysis of tree rings to climate is usually based on climate data on a monthly timescale, but the correlation between tree-ring chronology and climate on the pentad (five days) timescale is more significant and better reflects the true climate signals recorded in tree rings [31]. Therefore, we calculated the correlation between tree-ring δ¹⁸O and precipitation, temperature, and RH at the pentad scale. The season selected for climate response analysis extends from the twentieth pentad (early April) of the preceding year to the sixtieth pentad (late October) of the current year, considering the influence of the previous year's climate on tree growth (Figure 4). The correlation between tree-ring δ¹⁸O and temperature, precipitation, and RH varied significantly in each pentad, with strong volatility (Figure 4). Tree-ring δ¹⁸O was mainly positively correlated with temperature, with a significant correlation occurring in the twenty-third and twenty-ninth pentads of the current year. There was mainly a positive correlation between tree-ring δ¹⁸O and precipitation, and the significantly correlated seasons included some pentads in the previous year and the current year (such as the 21st, 43rd, 44th, 47th, and 48th pentads of the previous year, and the 3rd, 21st, 28th, 38th, 40th, and 46th pentads of the current year).

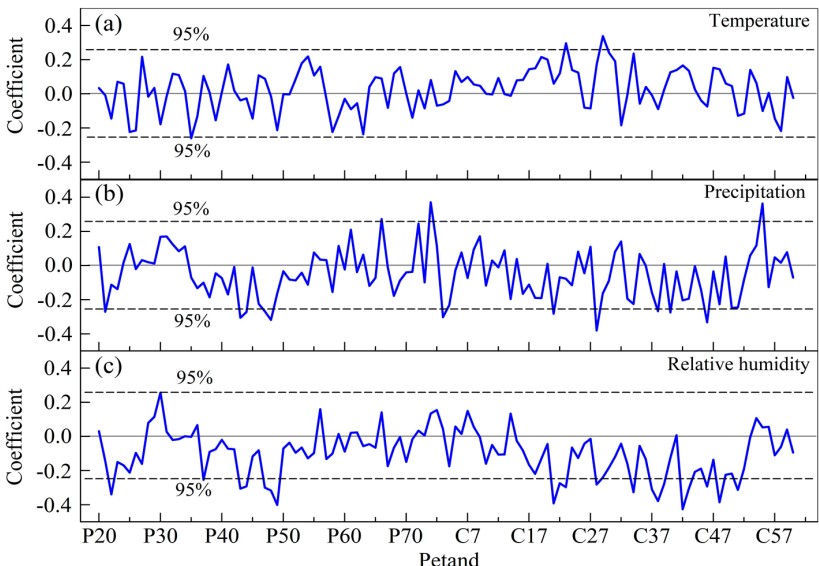

**Figure 4.** Correlation coefficients between tree-ring δ¹⁸O and each climate element on a pentad (five days) timescale from 1960 to 2016. The 95% confidence level is represented by the black dashed lines. P and C are used to denote the previous and current year, respectively.

By analyzing the pentad meteorological data from every season, we found that the tree-ring δ¹⁸O had the highest correlation with temperature during the 29th to 31st pentad of the current year (r = 0.350, $p < 0.01$), and that the strongest association with precipitation appeared in the season from the 41st pentad of the previous year to the 40th pentad of the current year (r = −0.625, $p < 0.001$). Lastly, the highest correlation with RH was found in the season from the 21st to 52nd pentad of the current year (r = −0.556, $p < 0.001$). It can be seen that precipitation had the most significant impact on tree-ring δ¹⁸O, which indicated that the YMMs tree-ring δ¹⁸O mainly reflected precipitation change. The period of strongest connection between tree-ring δ¹⁸O and precipitation occurred from July 21 of the previous year to July 20 of the current year. This time span was approximately equivalent to the period from August of the previous year to July of the current year. Therefore, we calculated the correlation between tree-ring δ¹⁸O and precipitation from the previous August to the current July ($P_{A-J}$, r = −0.565, 1960–2016). This indicated that tree-ring δ¹⁸O could to some extent reflect $P_{A-J}$ changes. In general, when precipitation is high, the isotopic composition of rainwater is relatively depleted. Therefore, more precipitation is associated with more δ¹⁸O-depleted precipitation, which in turn causes tree-ring δ¹⁸O to be lower [32].

Conversely, because of insufficient atmospheric vapor pressure, low precipitation may lead to higher concentrations of $\delta^{18}O$ in leaf and tree rings [33]. In addition, the previous year's carbohydrates formed in trees make an important contribution to the formation of the current year's earlywood, while the latewood formation is mainly dependent on the carbohydrates of the current year [34]. Therefore, the YMMs tree-ring $\delta^{18}O$ reflects the precipitation changes from the previous and current growing seasons of *Pinus tabulaeformis*.

### 3.3. YMMs Precipitation Reconstruction

Based on the correlation between tree-ring $\delta^{18}O$ and precipitation from the 41st pentad of the previous year to the 40th pentad of the current year ($P_{41-40}$), we used the least squares linear regression to establish a transformation equation between them during their common period of 1960 to 2016: $P_{41-40} = 3026.13 - 80.83 \times \delta^{18}O_t$ ($n = 57$, r = 0.625, $R^2 = 0.391$, $R^2$adj = 0.380, F = 35.324, $p < 0.001$). The stability and reliability of the regression equation was verified using the split-sample method (Table 3). These validation trials were performed by dividing the common period (1960–2016) into two parts: 30 years for calibration (1960–1989/1987–2016), and the remaining 27 years for verification (1990–2016/1960–1986). The two rigorous verification statistics (RE and CE) were positive, indicating a rigorous model skill. All these statistical parameters show that the regression model used in our reconstruction was stable and reliable.

**Table 3.** The transfer function's split calibration–verification test for the period of 1960–2016.

| Calibration | | | | Verification | | | | | |
|---|---|---|---|---|---|---|---|---|---|
| **Period** | **r** | **R2** | **ST** | **Period** | **r** | **R2** | **RE** | **CE** | **ST** |
| 1960–1989 | 0.645 ** | 0.416 | 23 ** | 1990–2016 | 0.654 ** | 0.428 | 0.236 | 0.226 | 18 |
| 1987–2016 | 0.676 ** | 0.457 | 22 * | 1960–1986 | 0.598 ** | 0.358 | 0.310 | 0.242 | 21 ** |
| 1960–2016 | 0.625 ** | 0.391 | 40 ** | | | | | | |

** $p < 0.01$, * $p < 0.05$.

For the modeling period of 1960–2016, the reconstructed precipitation was generally consistent with the observations (Figure 5a), and the first-order differences in the reconstructed and observed precipitation showed similar variations (Figure 5b). The correlation between the first-order differences of the reconstructed and observed precipitation was 0.564 ($p < 0.001$), which indicates that our reconstruction captured the changes characteristic of the observed precipitation at both high and low frequencies.

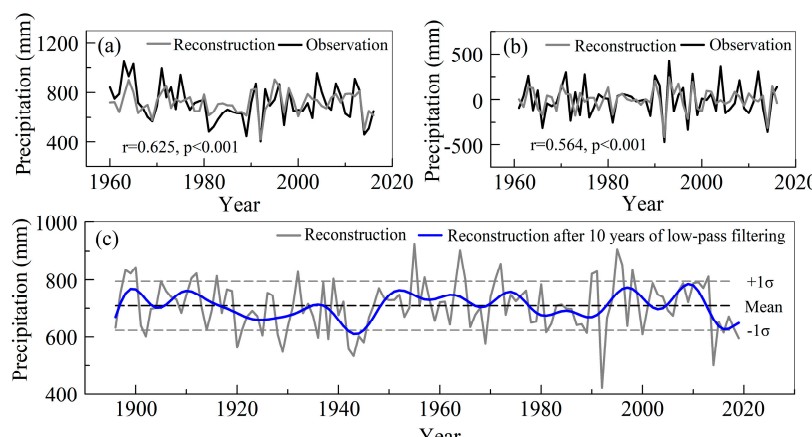

**Figure 5.** Observed and reconstructed precipitation of the YMMs. (**a**) Comparisons between the reconstructed and observed precipitation for 1960–2016 and (**b**) their first-order differences. (**c**) Precipitation reconstruction from 1896 to 2019. In (**c**), the blue curve represents the reconstruction after 10 years of low-pass filtering, and the horizontal lines indicate the overall mean of the reconstruction and one deviation (σ) from the mean value.

*3.4. YMMs Precipitation Changes over the Past 124 Years*

According to the length of tree-ring $\delta^{18}O$ sequence, we reconstructed the $P_{41-40}$ change from 1896 to 2019 (Figure 5c). During the past 124 years, the average precipitation in the YMMs was 708.7 mm, with a standard deviation (σ) of 85.9 mm. Based on the relationship between reconstructed value and the average plus or minus one σ, we considered wet years to be defined as years with precipitation greater than 794.6 mm and dry years as years with precipitation less than 622.8 mm. A total of 19 dry years and 20 wet years were found, with the rest being normal years (Figure 5c, Table 4). Most wet and dry years could be confirmed by historical records [35] and a database of the study region (http://lib.sdsqw.cn/ftr/ftr.htm, accessed on 23 December 2023).

**Table 4.** Dry/wet years in the reconstructed precipitation of the YMMs from 1896 to 2019 and the corresponding descriptions in the historical document "Local Chronicles of Linyi" [35] and in the database "Situation of Shandong Province" (http://lib.sdsqw.cn/ftr/ftr.htm, accessed on 23 December 2023).

| Wet Years | Documentary Records | Precipitation (mm) | Dry Years | Documentary Records | Precipitation(mm) |
|---|---|---|---|---|---|
| 1898 | NA * | 833.8 | 1902 | NA | 601.5 |
| 1899 | NA | 822.1 | 1920 | NA | 562.0 |
| 1900 | Heavy rain in Linyi damaged crops. | 844.1 | 1925 | No rain from March to middle of July in Juxian County. | 605.1 |
| 1911 | Floods caused disasters in six counties of Linyi. | 805.4 | 1928 | Severe drought hit Linyi, resulting in an average crop loss of 30%. | 609.2 |
| 1912 | NA | 822.1 | 1929 | NA | 547.6 |
| 1916 | Linyi suffered from heavy rains, which reduced crop yields. | 813.1 | 1936 | NA | 596.9 |
| 1932 | Heavy rain in Linyi led to flash floods breaking out. | 827.6 | 1938 | NA | 592.3 |
| 1939 | NA | 803.8 | 1942 | Feinan and Feibei Counties were hit by drought, with rivers drying up and crops dying. | 557.1 |
| 1955 | River banks burst in Juxian County after heavy rains, injuring people. | 923.6 | 1943 | Tancheng County experienced severe drought. | 532.4 |
| 1957 | Linyi was subjected to persistent heavy rains and concentrated rains, causing flash floods. | 808.1 | 1944 | Linyi experienced severe drought, with almost six months of little to no rainfall, resulting in near-total crop failure. | 602.3 |
| 1964 | Linyi experienced floods due to excessive rainfall. | 902.4 | 1945 | NA | 580.7 |

**Table 4.** *Cont.*

| Wet Years | Documentary Records | Precipitation (mm) | Dry Years | Documentary Records | Precipitation(mm) |
|---|---|---|---|---|---|
| 1965 | Feixian County was soaked by heavy rains, flash floods broke out, and rivers overflowed. | 802.9 | 1969 | Mengyin and Yiyuan Counties experienced severe drought, resulting in a reduction of over 30% in crop yields. | 577.6 |
| 1971 | Linyi experienced summer floods. | 803.5 | 1981 | 300,000 people in Linyi had difficulty accessing drinking water due to drought. | 616.2 |
| 1972 | NA | 856.1 | 1989 | There was an unprecedented drought in Linyi, with rivers drying up and groundwater levels declining. | 613.7 |
| 1990 | Heavy rains in Linyi caused reservoirs to overflow and river floods to surge. | 820.4 | 1992 | Linyi suffered from severe drought, unprecedented in nearly a century, leading to the drying up of rivers. | 421.0 |
| 1991 | Linyi suffered the largest rainstorm since 1974, causing massive economic losses. | 829.0 | 2000 | 57% of the arable land in Linyi encountered severe drought. | 609.2 |
| 1995 | Heavy rain hits Linyi. | 905.8 | 2014 | Linyi suffered from severe drought, with precipitation nearly 62% less than the same period in previous years. | 501.4 |
| 1996 | NA | 851.4 | 2016 | Linyi suffered from severe drought, with precipitation nearly 50% less than the same period in previous years. | 615.2 |
| 1998 | NA | 834.4 | 2019 | Precipitation in Pingyi and Mengyin Counties decreased nearly 50% compared with the same period in previous years. | 595.3 |
| 2013 | NA | 811.3 | | | |

* NA: not available.

### 3.5. Spatial Representativeness of the Reconstructed Precipitation

We found that the reconstructed $P_{41-40}$ can to some extent reflect the $P_{A-J}$ changes. Therefore, we calculated the spatial correlation between the observed and reconstructed $P_{41-40}$ and CRU gridded $P_{A-J}$ (1960–2016). The spatial correlation pattern between observed $P_{41-40}$ and CRU precipitation was consistent with that between the reconstructed $P_{41-40}$ and CRU precipitation (Figure 6a,b), which shows that our reconstruction mainly reflected the precipitation changes in Shandong Province. The correlation between reconstructed precipitation and CRU $P_{A-J}$ from 1902 to 2019 indicates (Figure 6c) that our reconstruction could also, to some extent, reflect regional precipitation changes over the past 100 years.

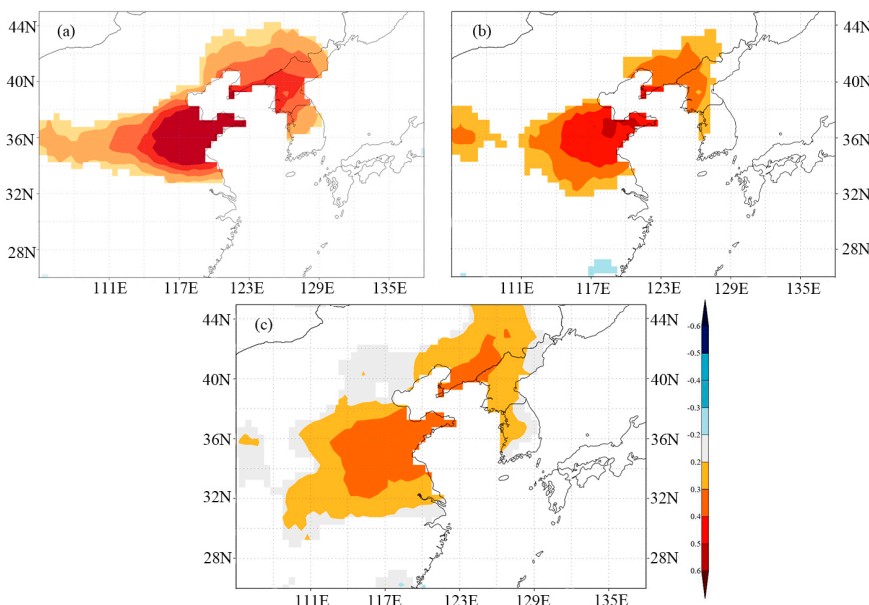

**Figure 6.** Spatial correlations of the gridded precipitation from the previous August to current July ($P_{A-J}$) from CRU with the observed $P_{41-40}$ (**a**) and reconstructed $P_{41-40}$ (**b**) for 1960–2016. (**c**) Correlation between the gridded $P_{A-J}$ and reconstructed $P_{41-40}$ for 1902–2019.

We compared the reconstructed precipitation with the CRU grid precipitation in the central part of eastern China (33–38° N, 115–123° E) and found that there was a significant relationship between them (Figure 7). The correlation between them was 0.519 ($p < 0.001$) from 1960 to 2019 and 0.438 ($p < 0.001$) from 1902 to 2019. In addition, the correlation between them for the period 1902–1959 was also significant ($r = 0.323$, $p = 0.013$). This further demonstrates that our reconstructed precipitation based on tree-ring $\delta^{18}O$ represents the precipitation changes in Shandong Province and the central part of eastern China.

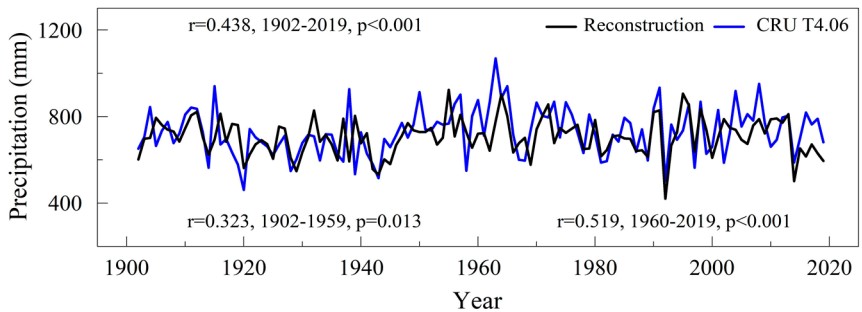

**Figure 7.** Comparison of the relationship between the reconstructed $P_{41-40}$ (black line) and reanalyzed gridded CRU $P_{A-J}$ (blue line).

### 3.6. Comparisons between YMMs Tree-Ring $\delta^{18}O$ Series and Other $\delta^{18}O$ Records

To further verify the reliability of our tree-ring $\delta^{18}O$ series, we compared it with tree-ring $\delta^{18}O$ data from Mt. Tianmu (MT) in eastern China [18] and Jirisan National Park (JNP) in southern South Korea [36] (Figures 1 and 8). It was found that there was a consistent change pattern between the tree-ring $\delta^{18}O$ of YMMs and that of MT. The tree-ring $\delta^{18}O$ of MT had the most significant correlation with the June–October PDSI ($r = -0.636$, $p < 0.001$) and the June–September precipitation ($r = -0.611$, $p < 0.001$) [18]. This indicates that the MT tree-ring $\delta^{18}O$ mainly reflected PDSI changes, but precipitation also had a significant impact on tree-ring $\delta^{18}O$ changes. Furthermore, a notable association was found between the tree-ring $\delta^{18}O$ series of the YMMs and JNP in southern South Korea, exhibiting comparable variations in both high and low frequencies (Figure 8). The JNP tree-ring $\delta^{18}O$ was also

negatively correlated with precipitation and it could be used as a proxy for precipitation change in May–July in southern South Korea [36].

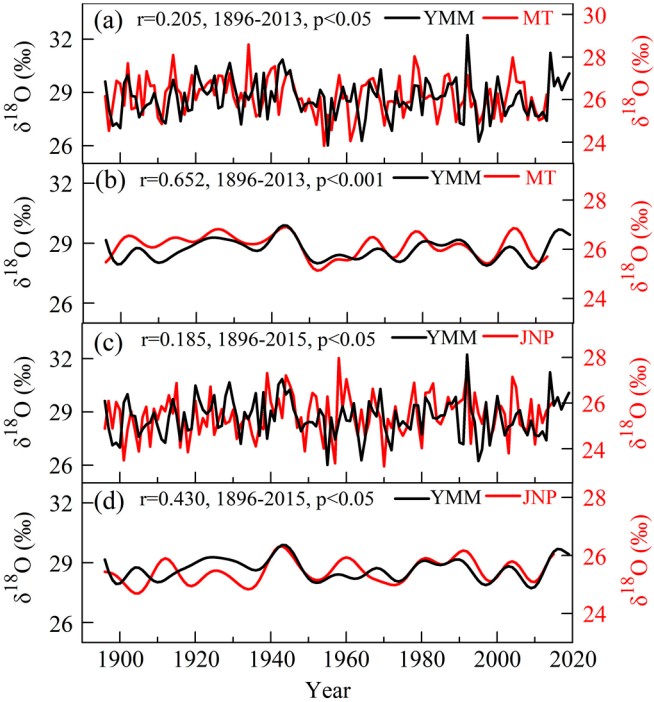

**Figure 8.** Comparisons between the YMMs tree-ring $\delta^{18}$O series in this study, (**a**,**b**) tree-ring $\delta^{18}$O data from Mt. Tianmu (MT) in eastern China [18], and (**c**,**d**) tree-ring $\delta^{18}$O values from Jirisan National Park (JNP) in southern South Korea [36]. (**a**,**c**) are original series, and (**b**,**d**) are 10-year low-pass filter series.

Our findings indicate that there was no significant correlation between the tree-ring $\delta^{18}$O of Nagano and Shiga (NS) in central Japan [37] and that of the Ordos Plateau (OP) in northwestern China [38] at high frequencies (Figure 1). However, there was some correlation at low frequencies. Specifically, after applying a low-pass filter for ten years, the correlation coefficient between the tree-ring $\delta^{18}$O sequences of the YMMs and NS was 0.281 (1896–2005), while the correlation coefficient between the tree-ring $\delta^{18}$O sequences of YMMs and OP was 0.208 (1896–2012). This also suggested that tree-ring $\delta^{18}$O at a given site might reflect large-scale changes in $\delta^{18}$O at low frequencies. It should be noted that tree-ring $\delta^{18}$O in both the OP and NS had more relative humidity signals than precipitation [37,38]. The OP tree-ring $\delta^{18}$O reflected, to some extent, the changes in the Asian summer monsoon (ASM) [38], and the precipitation in the YMMs is also influenced by the ASM. Thus, there was some relationship between tree-ring $\delta^{18}$O of the YMMs and OP. The tree-ring $\delta^{18}$O in JNP in southern South Korea was similarly influenced by the rainfall in Japan [37], leading to a close relationship between the tree-ring $\delta^{18}$O of the YMMs and JNP. Therefore, the tree-ring $\delta^{18}$O of the YMMs has some correspondence with that of NS in central Japan.

### 3.7. Possible Driving Mechanism of the YMMs Precipitation

The spatial correlation between the HadISST (Hadley Centre Sea Ice and Sea Surface Temperature) grid data 39] and the observed precipitation as well as the reconstructed precipitation showed that SST in the North Pacific had some impact on precipitation change in the YMMs. Not only was the correlation pattern between the observed precipitation and SST similar to that between of the reconstructed precipitation and SST for 1960–2016, but the correlation between the reconstructed precipitation and SST was also significant for 1896–2019. The significant correlation in the North Pacific (Figure 9) indicates that the PDO (Pacific decadal oscillation) was a major influencing factor of precipitation changes

in the YMMs. Therefore, we further analyzed the relationship between the reconstructed precipitation and the PDO.

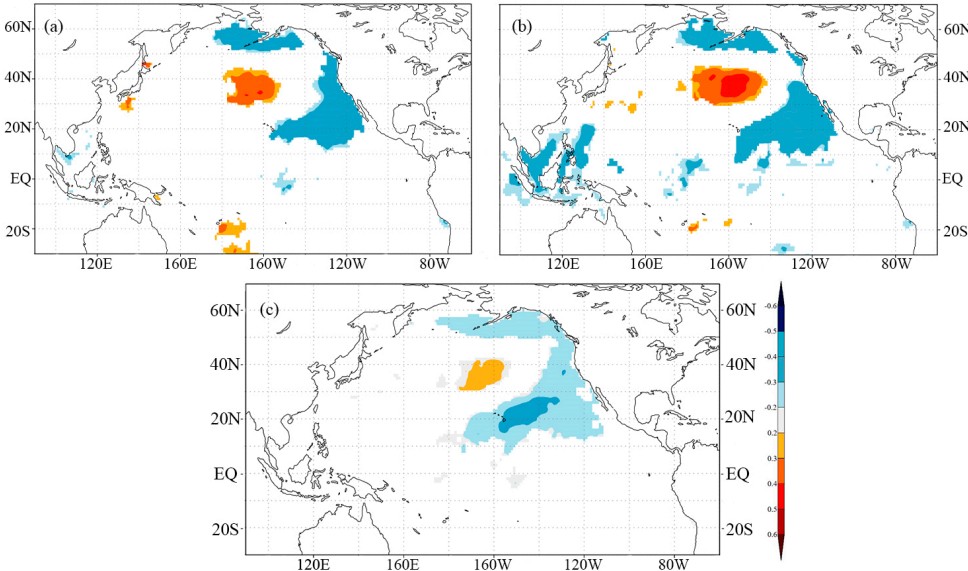

**Figure 9.** Spatial correlation between the HadISST grid data [39], (**a**) observed precipitation and (**b**) our reconstructed precipitation during the period of 1960–2016. (**c**) Correlation between the gridded SST and reconstructed precipitation during the period of 1896–2019.

The relationships between three sets of PDO reanalysis data [40–42] and precipitation were analyzed (Figure 10). Both at high and low frequencies, each set of PDO indices was significantly correlated with reconstructed precipitation. This suggests that the PDO exerted a notable influence on the changes in YMM precipitation. The PDO influences precipitation change in the central part of eastern China by modulating the location of the subtropical high and the strength of the Asian summer monsoon [43]. When the PDO is in the warm phase, the western Pacific subtropical high moves southward, the East Asian summer monsoon weakens, and the trade wind in the tropical Pacific also weakens. As a result, there is significantly less water vapor transported from the western Pacific to the central region of eastern China [44].

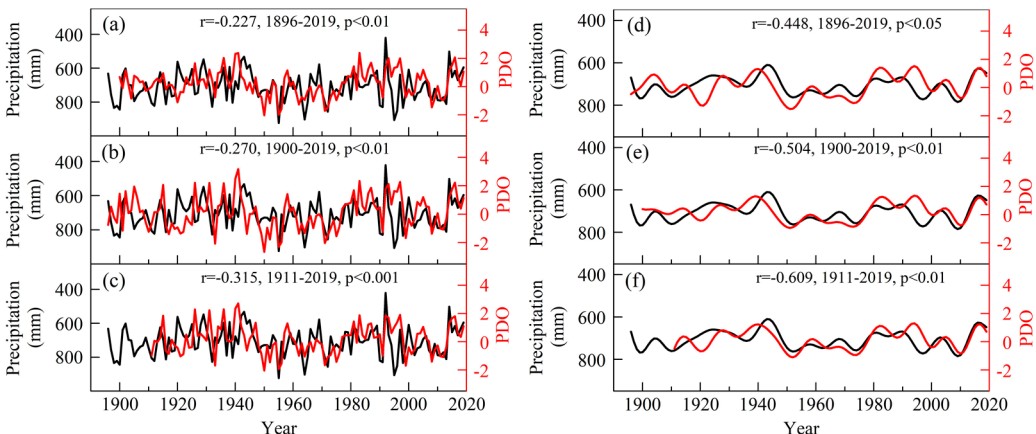

**Figure 10.** Comparisons between the reconstructed precipitation in the YMMs, (**a**) the PDO from NOAA based on the ERSST [40], (**b**) the PDO from SWFSC (Southwest Fisheries Science Center) [41], and (**c**) the PDO from the KNMI Climate Explorer based on the HadISST [42]. (**d**–**f**) are the precipitation and PDO after applying a 10-year low-pass filter. The PDO is the monthly mean Pacific Decadal Oscillation (PDO) index from April to July.

## 4. Conclusions

One tree-ring $\delta^{18}$O series spanning 124 years was carried out with tree-ring cores of *Pinus tabulaeformis* from the YMMs in the central region of eastern China. After climate response analysis, it was found that tree-ring $\delta^{18}$O was majorly influenced by $P_{41-40}$ change. Therefore, the YMMs $P_{41-40}$ from 1896 to 2019 was reconstructed. The reconstruction could explain 39% of the variance of observed $P_{41-40}$ during their common period of 1960–2016. There were 19 dry years, 20 wet years and 85 normal years over the past 124 years. The spatial correlation indicates the $P_{41-40}$ reconstruction reflects precipitation changes in Shandong Province in eastern China. The YMMs tree-ring $\delta^{18}$O showed a significant correlation with tree-ring $\delta^{18}$O in nearby regions, such as the MT in eastern China and JNP in southern South Korea, at both high and low frequencies. However, the YMMs tree-ring $\delta^{18}$O showed only a limited correlation with tree-ring $\delta^{18}$O from far regions, such as NS in central Japan and OP in northwestern China, at low frequencies. In addition, the PDO had a significant impact on the changes in YMMs precipitation and tree-ring $\delta^{18}$O.

**Author Contributions:** Conceptualization, C.S., Q.L. and Y.L.; methodology, C.S., X.W. and Q.L.; investigation, C.S. and M.R.; data curation, Y.L., M.R., Q.C. and H.S.; writing—original draft and revising, C.S., X.W. and Q.L.; visualization, X.W. and Y.M. All authors have read and agreed to the published version of the manuscript.

**Funding:** This study was jointly supported by grants from the National Natural Science Foundation of China (41902189; 42173080), the Chinese Academy of Sciences (XDB40000000), the CAS 'Light of West China' Program (XAB2021YN04), and the State Key Laboratory of Loess and Quaternary Geology, Institute of Earth Environment, CAS (SKLLQG1714).

**Data Availability Statement:** The meteorological data mentioned in this study can be accessed through the public Internet.

**Conflicts of Interest:** The authors declare no conflicts of interest.

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
