# Peer review of "Tree-Ring Stable Oxygen Isotope Ratio (δ18O) Records Precipitation Changes over the past Century in the Central Part of Eastern China"

_forests, doi:10.3390/f15010128_

Round 1

Reviewer 1 Report

Comments and Suggestions for Authors

Author Response

Below are some general and specific comments on the paper entitled “Tree-ring stable oxygen isotope ratio (δ18O) change in the central part of eastern China over the past century”. (Manuscript Number: forests-2783002). This paper is interesting because it presents new data on tree-ring oxygen isotopes of the central part of eastern China to evaluate the rainfall history and the connection with Pacific decadal oscillation. However, the manuscript is not particularly clear in a few points and the following suggested changes may rectify this problem.

[Reply] Thank you for your positive comments.

General comments:

  1. In the section of 2, the author should describe the general information of tree spices, such as the season when annual rings are formed. In addition, cross-dating results with ring width and the climate response should be explained.

[Reply] The general information of tree spices and the cross-dating results with ring width were added. And the climate response was explained. Please see Line 108-109, Line 124-127 and Line 228-237.

  1. In the section of 3.6, the authors compared between YMM and other sites. It would be easier for the readers to understand if a detailed comparison included, citing the climatic response of annual rings in other regions.

[Reply] The detailed comparison was added with citing the climatic response of tree-ring isotope in other regions. Please Line 307-308, Line 312-314 and Line 327-329.

Specific comments:

Table 2: Add the formation duration (e.g. year-year) of each tree-ring sample.

[Reply] The formation duration (e.g. year-year) was added. Please see new Table 1.

Table 2: Add the meaning of AR1, citing literature if available.

[Reply] The meaning of AR1 was added and a literature was cited. Please see new Table 1.

Reviewer 2 Report

Comments and Suggestions for Authors

The manuscript (MS) is dedicated to developing of tools allowing one to reconstruct weather parameters from stable isotopes in tree rings. Overall, the research is good planed and clearly presented. Still, minor improvements may be desirable before the publication.

Scientifically, there is a question regarding the reconstruction made and its independent verification. Table 4 gives the data on reconstructed years and classifying them as dry and wet. Why not verify the information from independent sources? Say, there are 4 years in a row 1942-1945 that are said to be dry. There must be historical records confirming that these years were really dry-from agriculture data or whatever. Recent dry years (like 1992) could also be confirmed from historical data.

Is that right that you used only 5 cores from 98 cores taken? What is the sense then to take so many cores?

Minor comments

l. 18, 64 -> the Latin name of species should be given together with the author names at the first appearance in the text.

l. 30 -> 'where' doesn't seem to be the right word here.

l. 48 -> has?

l. 56, 61, 65, 68 -> please, pay attention to citing format.

l. 201 -> Pentad?

Comments on the Quality of English Language

English is clear enough, the MS is good written, the language is above the average for non-native speakers as far as I can judge. Minor editing only might be required.

Author Response

The manuscript (MS) is dedicated to developing of tools allowing one to reconstruct weather parameters from stable isotopes in tree rings. Overall, the research is good planed and clearly presented. Still, minor improvements may be desirable before the publication.

[Reply] Thank you for your positive comments.

Scientifically, there is a question regarding the reconstruction made and its independent verification. Table 4 gives the data on reconstructed years and classifying them as dry and wet. Why not verify the information from independent sources? Say, there are 4 years in a row 1942-1945 that are said to be dry. There must be historical records confirming that these years were really dry-from agriculture data or whatever. Recent dry years (like 1992) could also be confirmed from historical data.

[Reply] The reconstructed series was verified by reanalysis data from the Climatic Research Unit (CRU). In addition, the dry and wet years classified were also confirmed by historical records and database of the study region. Please see Line 271-276, new Table 4 and Figure 7.

Is that right that you used only 5 cores from 98 cores taken? What is the sense then to take so many cores?

[Reply] Because we also conducted tree-ring width research, 98 cores were taken. 5 cores were selected to do tree-ring isotope research in this study. The research about tree-ring width is still in progress.

Minor comments

  1. 18, 64 -> the Latin name of species should be given together with the author names at the first appearance in the text.

[Reply] It was changed. Please see Line 18 and Line 65.

  1. 30 -> 'where' doesn't seem to be the right word here.

[Reply] It was changed. Please see Line 30.

  1. 48 -> has?

[Reply] It was changed. Please see Line 48.

  1. 56, 61, 65, 68 -> please, pay attention to citing format.

[Reply] It was changed. Please see Line 56-72.

  1. 201 -> Pentad?

[Reply] It was changed. Please see Line 212.

Comments on the Quality of English Language

English is clear enough, the MS is good written, the language is above the average for non-native speakers as far as I can judge. Minor editing only might be required.

[Reply] The English was improved.